# Structural delineation of potent transmission-blocking epitope I on malaria antigen Pfs48/45

Prasun Kundu[1], Anthony Semesi[1], Matthijs M. Jore[2], Merribeth J. Morin[3], Virginia L. Price[3], Alice Liang[4], Jingxing Li[4], Kazutoyo Miura[5], Robert W. Sauerwein[2], C. Richter King[3] & Jean-Philippe Julien [1,6]

Interventions that can block the transmission of malaria-causing *Plasmodium falciparum* (Pf) between the human host and Anopheles vector have the potential to reduce the incidence of malaria. Pfs48/45 is a gametocyte surface protein critical for parasite development and transmission, and its targeting by monoclonal antibody (mAb) 85RF45.1 leads to the potent reduction of parasite transmission. Here, we reveal how the Pfs48/45 6C domain adopts a (SAG1)-related-sequence (SRS) fold. We structurally delineate potent epitope I and show how mAb 85RF45.1 recognizes an electronegative surface with nanomolar affinity. Analysis of Pfs48/45 sequences reveals that polymorphisms are rare for residues involved at the binding interface. Humanization of rat-derived mAb 85RF45.1 conserved the mode of recognition and activity of the parental antibody, while also improving its thermostability. Our work has implications for the development of transmission-blocking interventions, both through improving vaccine designs and the testing of passive delivery of mAbs in humans.

---

[1] Program in Molecular Medicine, The Hospital for Sick Children Research Institute, Toronto M5G 0A4 ON, Canada. [2] Department of Medical Microbiology, Radboud University Medical Center, Nijmegen 6500 HB, Netherlands. [3] PATH's Malaria Vaccine Initiative, Washington 20001 DC, USA. [4] LakePharma Inc., Belmont 94002 CA, USA. [5] Laboratory of Malaria and Vector Research, National Institute of Allergy and Infectious Diseases, National Institutes of Health, Rockville 20852 MD, USA. [6] Departments of Biochemistry and Immunology, University of Toronto, Toronto M5S 1A8 ON, Canada. These authors contributed equally: Prasun Kundu, Anthony Semesi. Correspondence and requests for materials should be addressed to J.-P.J. (email: jean-philippe.julien@sickkids.ca)

M alaria is a global health priority, with an estimated 216 million cases worldwide in 2016 alone[1]. The *Plasmodium falciparum* (Pf) parasite is responsible for most malaria-related mortalities, with over two-thirds occurring in children under 5 years of age. Disrupting the Pf life cycle as the parasite circulates between humans and Anopheles mosquitoes has the potential to reduce infections within communities, and thus reduce illness and death[2]. It was recognized decades ago that antibodies have the potential to inhibit onward transmission of the parasite when passed by malaria-infected humans to the *Anopheles* mosquito vector[3–5]. Transmission-blocking vaccines (TBVs) are based on this principle, and aim to elicit antibodies in humans that can reduce Pf transmission to the vector when mosquitoes ingest these antibodies during feeding[6]. Target proteins for TBV development are located on the surface of gametocytes/gametes (P48/45[7], P230[8]) and zygotes/ookinetes (P25, P28)[9,10], or are expressed within the mosquito midgut (e.g. APN1[11] and FREP1[12]).

Pfs48/45 is a cysteine-rich surface protein that plays a critical role in male gamete fertility[13]. Antibodies against Pfs48/45 have been shown to prevent parasite development and transmission[14,15]. Because Pfs48/45 is located on gametocytes/ gametes that circulate in humans, naturally occurring antibodies against Pfs48/45 can be elicited in individuals living in malaria-endemic areas, and these antibodies have been demonstrated to possess transmission-blocking activity[16–21].

Pfs48/45 is organized into three domains, and epitope mapping has identified at least five predominant sites of antibody recognition that span its primary sequence[22,23] (Fig. 1a). Human, murine, and rat monoclonal antibodies (mAbs) have been derived that target epitopes I, IIb, III, and V[5,24–26]. Of these, mAb 85RF45.1, derived from rats immunized with whole gametocytes followed by fusion and screening by enzyme-linked immunosorbent assay for Pfs48/45 antibodies, was shown to be the most potent at inhibiting parasite transmission to *Anopheles* mosquitoes[24]. mAb 85RF45.1 is a rat IgG1 that binds to epitope I on Pfs48/45[24].

A three-dimensional structure of Pfs48/45 remains elusive. Such information would provide useful in devising strategies for improved recombinant expression of immunogens; indeed, monodisperse Pfs48/45 has been challenging to obtain and development of homogeneous and stable recombinant constructs is an area of intense research[7,25,27–30]. A molecular understanding of mAb 85RF45.1 binding to Pfs48/45 would also provide the blueprints for the development of next-generation Pfs48/45 immunogens that optimally present this potent epitope to be tested as TBVs. In addition, molecular details of this antibody–antigen interaction could facilitate the development of new antibody interventions. Here, we present the three-dimensional structure of the Pfs48/45 6C domain and delineate the potent 85RF45.1 epitope structurally. We also provide molecular details for the development of mAb TB31F, a humanized version of rat mAb 85RF45.1 with improved biophysical properties, now undergoing preclinical development.

## Results

**mAb 85RF45.1 recognizes a conformational epitope on Pfs48/45**. To uncover the potent epitope on Pfs48/45 bound by transmission-blocking mAb 85RF45.1, we solved the co-crystal structure of the 85RF45.1 Fab-Pfs48/45 6C complex to 2.7 Å resolution (Fig. 1b and Supplementary Table 1). All six complementarity determining regions (CDRs) are involved in mediating interactions with a conformational epitope on Pfs48/45 (Fig. 1c). The buried surface area of mAb 85RF45.1 on its antigen is extensive (1039 Å²), and primarily contributed by the heavy chain (650 Å²), and less by the light chain (389 Å²) (Supplementary Table 2). Ten discontinuous residues on Pfs48/45 are contacted by the antibody to form salt bridges or H-bonds, namely G346, D347, D351, Q355, E365, K394, K413, K414, D415, and K416 (Fig. 1c, insets). Our structural delineation of the antibody–antigen complex corroborates the importance of disulfide bonds and protein conformation for epitope I established by previous biochemical characterization[24].

**The Pfs48/45 6C domain adopts a SRS fold**. Analysis of the Pfs48/45 6C sequence reveals that proteins with the highest sequence identity include Pfs48/45 orthologs from *Plasmodium vivax*, *malariae*, and *ovale* (62–64%), as well as the gamete surface protein Pfs47 (27%) (Fig. 2a). Our crystal structure of Pfs48/45 6C now also allows for a complete analysis of its homology. Pfs48/45 6C contains a mix of parallel and anti-parallel ß-strands that organize in a five-on-four ß sandwich (a, a′, c, e, and f on b, d, g, and h), which is consistent with the (SAG1)-related-sequence (SRS) fold. The six cysteines in Pfs48/45 6C are paired in a C1–C2, C3–C6, C4–C5 disposition: C298–C327, C344–C412, and C352–C410 (Figs. 1a and 2b), as consistent with previous biochemical analysis of tryptic peptides by LC–MS/MS[30]. These disulfide bonds connect strands a and b, strands c and g, and loop d–d′ to strand g, respectively (Fig. 2b). The three proteins of known structure with highest structural similarity with Pfs48/45 6C are schizont surface protein Pf12 (Z-score 12.9), merozoite surface protein Pf41 (Z-score 12.0), and *Toxoplasma gondii* sporozoite-specific SAG protein (Z-score 10.5) (Fig. 2c)[31]. Pfs48/ 45 6C only shares 16–22% sequence identify with these proteins and consequently displays unique surface properties likely involved in male gamete fertility and giving the protein its distinct immunogenicity (Supplementary Figure 1).

**N-linked glycosylation improves expression of Pfs48/45 6C**. The Pfs48/45 6C construct used for structure determination was expressed in HEK293 mammalian cells, thus in a system capable of post-translational modifications. The primary amino acid sequence of Pfs48/45 6C encodes for two putative N-linked glycosylation sites when expressed in mammalian cells: N299 (NFS) and N303 (NVS), both located near the N terminus of the construct. Our crystal structure revealed that N303 is indeed glycosylated, and harbors a NAG residue residual from Endoglycosidase H (Endo H) treatment (Fig. 3a). In fact, NAG303 forms H-bonds with D390 and D391 of the neighboring loop and buries approximately 150 Å² of surface area on the two loops that connect strands a–a′ and strands e–f (Fig. 3a). On the other hand, N299, which immediately follows C298, is not glycosylated, and instead its sidechain plays an important structural role in the strand a-loop-strand a′ motif, mediating H-bonds with S301 (Fig. 3a). To determine whether N-linked glycosylation played a role in the expression of Pfs48/ 45 6C in our HEK293 system, we created single and double mutants removing the N-linked glycosylation sequon by introducing S301A and S305A mutations. Expression tests showed a substantial decrease in recovered protein levels when removing N303 glycosylation, with the double mutant yielding almost no Pfs48/45 6C (Fig. 3b). N-linked glycosylation was also critical to obtain a monomeric and monodisperse Pfs48/45 6C sample, as observed by size-exclusion chromatography (Fig. 3c). Importantly, the N303 N-linked glycan is distally located from the 85RF45.1 epitope. N-linked glycosylation resulting from expression in mammalian cells therefore favors production of well-folded Pfs48/45 6C, without affecting exposure of potent epitope I.

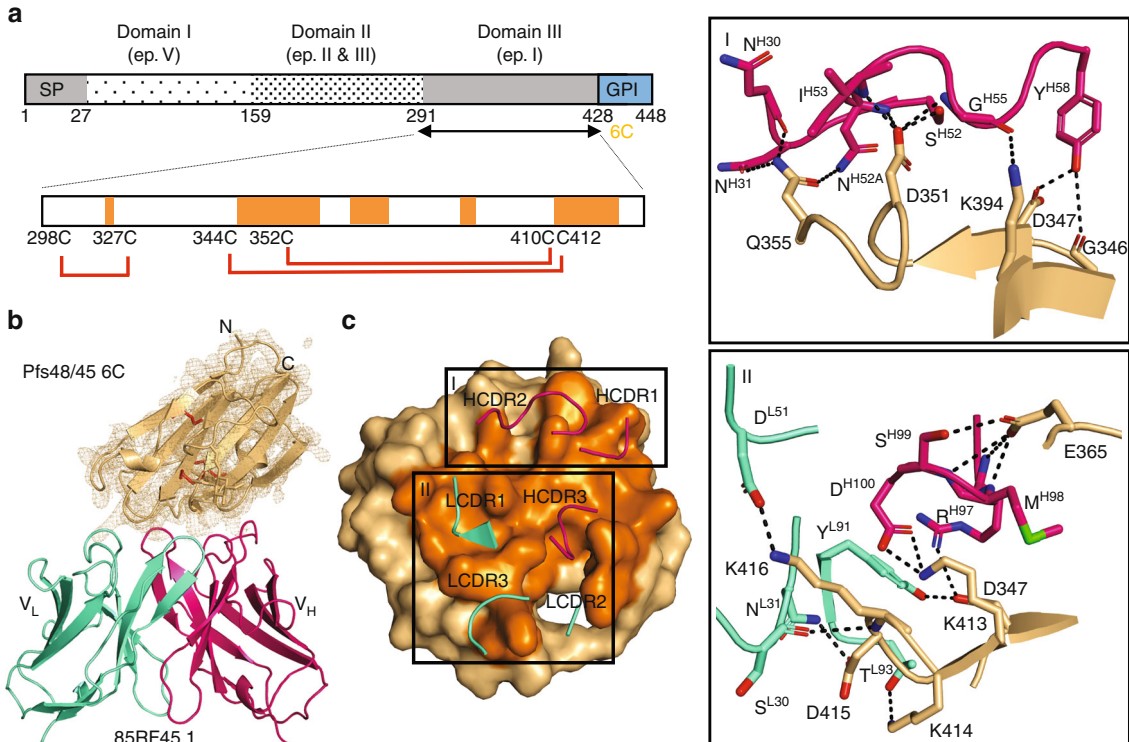

**Fig. 1** Recognition of Pfs48/45 6C by potent transmission-blocking mAb 85RF45.1. **a** Domain organization of Pfs48/45. In the zoom in view of the Pfs48/45 6C domain, the dark orange regions represent areas contacted by mAb 85RF45.1. Red lines indicate the disulfide bonding pattern. SP signal peptide, ep epitope, GPI glycosylphosphatidylinositol anchor. **b** Overview of the 85RF45.1 Fab-Pfs48/45 6C co-crystal structure rendered as secondary structure cartoon and surface. The orange mesh represents the composite omit electron density map rendered at a 1.2σ contour level around Pfs48/45 6C. The six-cysteine residues in Pfs48/45 6C are shown as red sticks. The 85RF45.1 light and heavy chains are colored teal and magenta, respectively. Only the Fab variable region is shown for clarity. **c** Residues of the 85RF45.1 epitope on Pfs48/45 (dark orange surface) are recognized by all six CDRs (secondary structure cartoon). The insets show residues involved in H-bonds and salt bridges (black dashes). Colors are as in **b**

**The 85RF45.1 epitope contains few sequence polymorphisms**. Next, we structurally mapped the sequence diversity documented for Pfs48/45. Deposited sequences of the Pfs48/45 6C domain from UniProt (28), NCBI (61), and PlasmoDB (195) revealed the following polymorphisms: V/D304, S/G313, L/I314, D/G315, S/N/C322, E/G333, I/V349, Q/L355, V/A356, A/T387, K/R404, K/E414, and T/I422 (Fig. 4a). The five Pfs48/45 residues predominantly involved in interacting with 85RF45.1 (D347, D351, K413, D415, and K416; all mediating extensive H-bonds/salt bridges and with individual BSA > 75 Å$^2$) are invariant (Fig. 4a). Two other Pfs48/45 residues mediating H-bonds with 85RF45.1 (E365 and K394) are also invariant (Fig. 4a). Nonetheless, three Pfs48/45 residues that are contacted by 85RF45.1 have documented polymorphisms (Fig. 4b): 349 is almost always Ile, but one sequence is reported with a Val at that position (AAL74379.1 [https://www.ncbi.nlm.nih.gov/protein/AAL74379.1])[32]; 355 is almost always Gln, but one sequence is reported with a Leu at that position (AAL74377.1 [https://www.ncbi.nlm.nih.gov/protein/AAL74377.1])[32]; 414 is almost always Lys, but one sequence is reported with a Glu at that position (AAL74361.1 [https://www.ncbi.nlm.nih.gov/protein/AAL74361.1])[32]. 85RF45.1 buries 36, 46, and 20 Å$^2$ of surface area on these three residues, respectively. Binding measurements of 85RF45.1 Fab to Pfs48/45 6C constructs possessing single point mutations that represent these polymorphisms revealed that the antibody is still capable of binding its epitope with nanomolar affinity (Fig. 4c). We conclude that the 85RF45.1 epitope is predominantly conserved, with rare polymorphisms located in secondary epitope regions (individual residues each represent <5% of total buried surface area).

**Humanization of mAb 85RF45.1**. Next, we sought to humanize mAb 85RF45.1, introducing as much human sequence as possible while retaining sequences that preserve the binding characteristics of the antibody. The closest human V germlines to the parental rat sequence were found to be IGLV6–57 (66.2% for the light chain) and IGHV3–7 (84.0% for the heavy chain). Three humanized light chains and three humanized heavy chains were designed, all exceeding our threshold humanness scores (Supplementary Figure 2). In all cases, CDRs remained unchanged. Three humanized light and heavy chains were combined to create a total of nine fully-humanized variant antibodies (Supplementary Figure 2). All three light chain combinations within the HC2 group bound the antigen similarly and the HC2 group affinities were higher than the HC1 and HC3 groups (Supplementary Figure 2). The HC2-LC1 (hereafter named TB31F, Fig. 5a) was selected for further characterization based on high binding activity and high human-like scores.

To characterize the ability of mAb TB31F to recognize Pfs48/45, we solved the crystal structure of TB31F Fab in complex with Pfs48/45 6C to 2.6 Å resolution (Fig. 5b and Supplementary Table 1). As expected from the preservation of CDR residues during humanization, mAb TB31F forms similar contacts with Pfs48/45 6C as mAb 85RF45.1 (Fig. 5b and Supplementary Table 2), with slight differences primarily attributable to flexible loops in Pfs48/45 6C and crystal packing contacts. Next, we investigated whether amino acid changes in the antibody framework regions during humanization impacted the structural disposition of CDRs. Crystal structures of unliganded 85RF45.1 Fab and unliganded TB31F Fab, solved to 3.15 and 1.5 Å

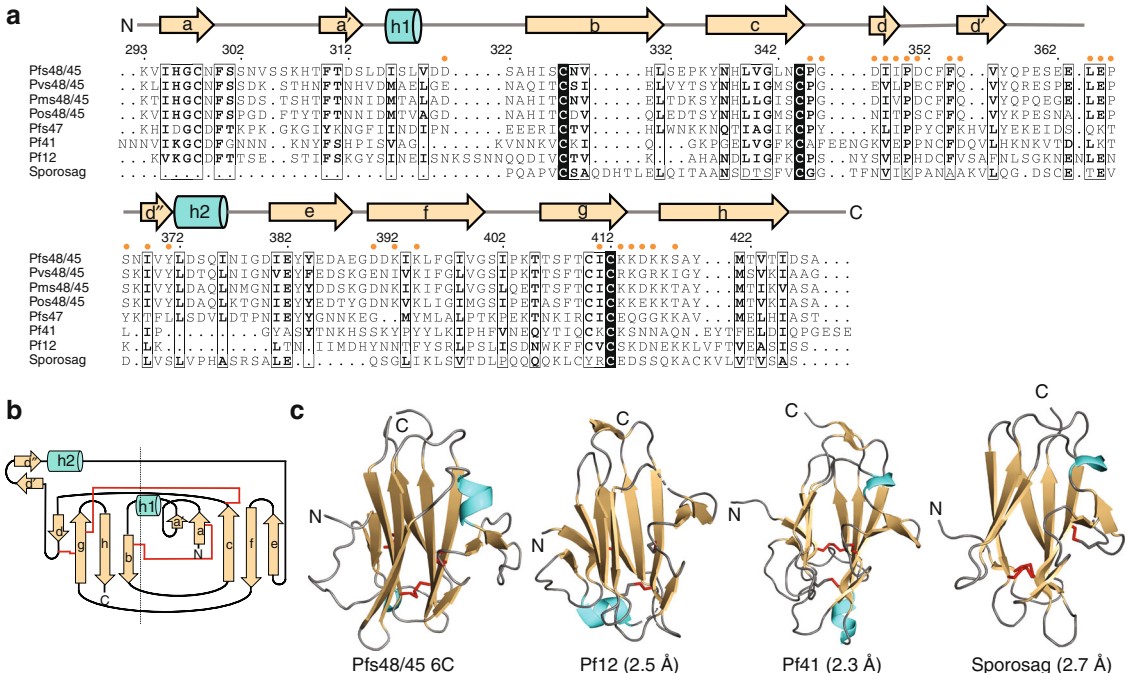

**Fig. 2** Sequence similarity and structural homology of Pfs48/45 6C. **a** Sequence alignment of Pfs48/45 6C with proteins of highest sequence identity (Pvs48/45, Pms48/45, Pos48/45, and Pf47) and structural homology (Pf12, Pf41, and Sporosag). Secondary structure elements of Pfs48/45 6C are shown atop the sequence alignment, with an arrow and a cylinder indicative of a strand and a helix, respectively. Orange dots above the Pfs48/45 sequence indicate residues contacted by mAb 85RF45.1. **b** Schematic diagram of Pfs48/45 showing a five-on-four ß sandwich disposition with parallel and anti-parallel ß-strands. Disulfide bonds are indicated by red lines. **c** Secondary structure cartoon representation of Pfs48/45 6C in comparison to known structures of highest homology (PDB IDs: 2YMO [https://www.rcsb.org/structure/2YMO][55], 4YS4 [https://www.rcsb.org/structure/4YS4][56], 2WNK [https://www.rcsb.org/structure/2WNK][57]). Disulfide bonds are represented as red sticks. Root mean square deviations (rmsd) between Pfs48/45 and the structures as calculated by the DALI server[31] are indicated in parentheses

resolutions, respectively, revealed a strikingly similar paratope configuration (Fig. 5c). The all-atom rmsd between the two unliganded variable domains is 0.88 Å, primarily arising from differences in loop conformations within the framework regions (Fig. 5c). These structures also revealed how the variable region is optimally configured for binding Pfs48/45 even when unliganded (all-atom rmsd 85RF45.1$_{bound/unbound}$: 1.52 Å; TB31F$_{bound/unbound}$: 1.25 Å). Despite this high structural similarity, amino acid changes during humanization resulted in electrostatic surfaces that are considerably different between the 85RF45.1 and TB31F variable regions, indicating the attainment of antibody resurfacing (Fig. 5d). Importantly, both 85RF45.1 Fab and TB31F Fab bound the antigen with nanomolar binding affinities (Fig. 5e and Supplementary Figure 3A) and the two mAbs inhibited 80% of oocyst intensity at similar concentrations of approximately 1 μg/mL (Fig. 5e, Supplementary Figure 4 and Supplementary Table 3). TB31F Fab also retained the ability of 85RF45.1 Fab to recognize Pfs48/45 with sequence polymorphisms in its epitope (Supplementary Figure 3B). In unfolding and aggregation experiments, TB31F Fab was found to have gained in thermostability, with a melting temperature ($T_m$) of 74.8 °C and an aggregation temperature ($T_{agg}$) of 73.0 °C, compared to 63.1 and 63.8 °C for 85RF45.1 Fab, respectively (Fig. 5e and Supplementary Figures 5 and 6). TB31F Fab shows thermostability throughout a wide pH range, whereas 85RF45.1 Fab is more dependent on pH (Supplementary Figures 5 and 6). These desirable biophysical properties for TB31F, combined with its retained nanomolar binding affinity and potent activity to inhibit parasite transmission, make this molecule an attractive biologic for malaria interventions.

## Discussion

Pfs48/45 is a prominent target for the development of biomedical interventions that can interrupt the transmission of Pf and contribute to the decrease of malaria incidence. Here, we report the three-dimensional structure of the Pfs48/45 6C domain and delineate the epitope targeted by one of the most potent transmission-blocking mAbs yet described, 85RF45.1.

Pfs48/45 6C adopts an SRS fold that consists of two β-sheets formed by a mix of parallel and anti-parallel β-strands. This fold has been proposed to derive from an ephrin-like precursor during evolution and is predicted to be adopted by at least 14 proteins in Pf[33,34]. Correspondingly, proteins of known structures that have the highest structural homology are parasitic proteins of this s48/45 family for which structures have been determined: Pf12 (Pf), Pf41 (Pf), and Sporosag (*Toxoplasma gondii*). It is thought that these glycosylphosphatidylinositol (GPI)-anchored parasitic proteins play an important role in cell adhesion and immune evasion, although no exact function has yet been assigned to Pf12 and Pf41[34]. Our structure also reveals that the electrostatic surface of Pfs48/45 6C is highly electronegative. Whether this property is associated with its function in male gamete fertility[13] and involved in ligand binding or scaffolding in a multi-protein complex[35] will be an important area of further studies. It will also be important to determine whether the potency of mAb 85RF45.1 is associated with disruption of the Pfs48/45 epitope I in protein–protein interactions.

To obtain the crystal structure of the 85RF45.1 Fab-Pfs48/45 6C complex, we expressed the antigen in mammalian cells, thus allowing for extensive post-translational modifications. The ability of Pf to carry out post-translational modifications and

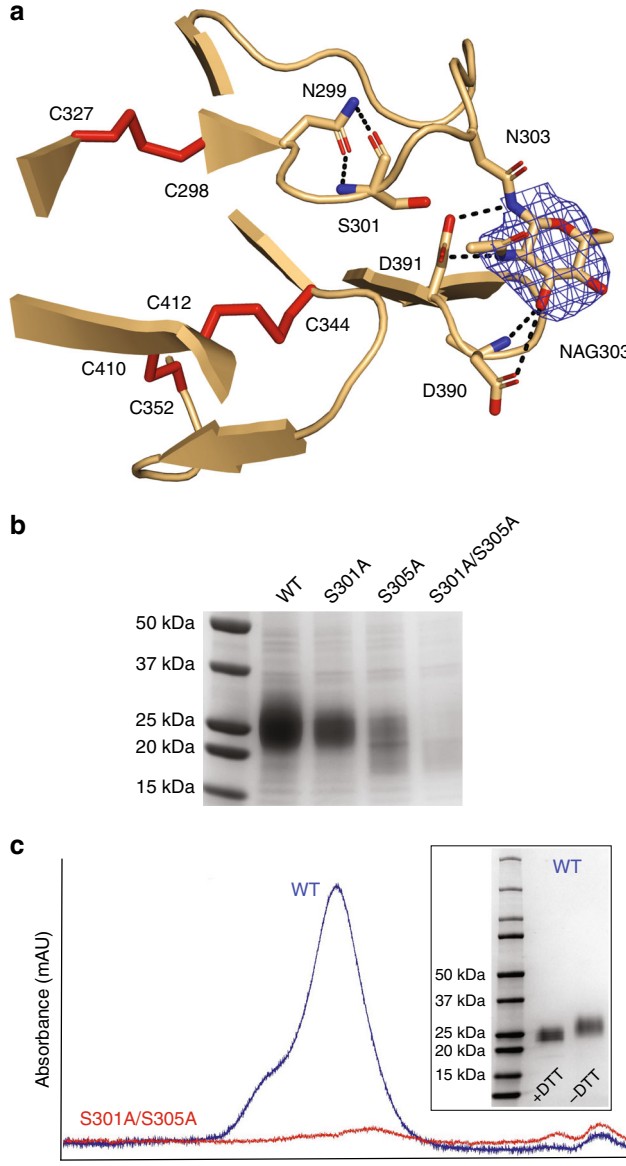

**Fig. 3** N-linked glycosylation improves expression of monodisperse Pfs48/45 6C. **a** The NAG at position N303 mediates H-bonds with D390 and D391, and helps stabilize the two loops that connect strands a–a' and strands e–f in the proximity of the three disulfide bonds (red sticks). The blue mesh is the composite omit electron density map rendered at a $1.2\sigma$ contour level around the N303 NAG residue. **b** SDS-PAGE under reducing conditions and stained with Coomassie Blue for recovered samples from Ni-NTA affinity chromatography with Pfs48/45 6C constructs of varying N-linked glycan content. **c** Size-exclusion chromatograms comparing the recovery and monodispersity of Pfs48/45 6C constructs that lack both (red), or no (blue) putative N-linked glycosylation sites. The inset is an SDS-PAGE gel stained with Coomassie Blue showing the purity of the monomeric, glycosylated Pfs48/45 6C construct

their exact composition is an area of active investigation[36–38], but it remains largely unclear how frequent N-linked glycosylation is on Pf surface proteins in the presence of NXS/T sequons[39]. Here, we found N-linked glycosylation to be critical for monodisperse Pfs48/45 6C expression. Accordingly, we observed in our Pfs48/45 6C crystal structures that a NAG residue at position 303 can bury >150 Å² of surface area between two loops and mediates

H-bonds with Pfs48/45 residues. Interestingly, in homologous merozoite protein Pf41, the location of the NAG residue in our Pfs48/45 6C structure is occupied by a helix from an upstream Pf41 N-terminal domain, which stacks against a similar groove. It is thus possible that in full-length Pfs48/45 elements located N-terminal of the Pfs48/45 6C fragment help stabilize the structure in a similar fashion. This hypothesis would be in agreement with the observation that refolding capacity was superior for a construct comprising both the middle four-cysteine domain and the C-terminal six-cysteine domain (10C)[15]. The success of fusing the N terminus of Pf glutamate-rich protein (PfGLURP R0 fragment) to express Pfs48/45 6C (R0.6C)[30] might also emanate from such favorable inter-domain stabilizing effects, possibly at the site occupied by the NAG residue here. Thus, our three-dimensional structures of Pfs48/45 6C now provide an opportunity for protein engineering to further stabilize the antigen, enhance its expression, and develop TBV immunogens that will optimally present epitope I to preferentially elicit potent transmission-blocking antibodies.

mAb 85RF45.1 is one of the most potent antibodies identified against any transmission-blocking target, with an IC$_{80}$ of approximately 1–3 µg/mL in standard membrane feeding assay (SMFA) experiments. We also now reveal that its epitope is largely conserved: mAb 85RF45.1 can accommodate all sequence polymorphisms previously reported and still bind with nanomolar affinity. However, whether this antibody can block field isolates containing these polymorphisms remains to be determined. mAb 85RF45.1 represents a unique tool to assess the efficacy of antibodies to reduce Pf transmission and malaria incidence in affected communities. For this purpose, optimizing the developability of this rat-originating mAb was an important objective. Here, we show how humanization of mAb 85RF45.1 into mAb TB31F retained the exact molecular mode of recognition of Pfs48/45, its high-affinity binding to the antigen, and its potent transmission-reducing activity in SMFA. In addition, humanized TB31F Fab shows improved thermostability characteristics with ~10 °C higher $T_m$ and $T_{agg}$ compared to its rat-derived counterpart. Thus, our data provide a salient example of how humanization can present an opportunity to simultaneously improve surface and biophysical profiles of non-human-derived mAbs, and will contribute to ongoing efforts to improve multi-objective antibody optimization functions[40,41].

mAb TB31F now offers a unique opportunity to test the efficacy of a Pfs48/45-based transmission-blocking intervention in humans. The mAb has entered good manufacturing production and preclinical safety testing. Phase I trials are planned to assess safety and tolerability. Pending a positive outcome, this will be followed by efficacy testing with the goal of establishing the level of mAb required to block parasite transmission from human to mosquito, as determined by direct skin feeding (infection prevalence endpoint). Comparison of these data to those from the laboratory-based SMFA, which employs an infection intensity (as opposed to infection prevalence) endpoint and is routinely used to inform go/no-go criteria in pre- and early-clinical testing, will enable "back-validation" of the SMFA to better inform its use as an early development stage-gate for measuring the efficacy of candidate vaccines and antibodies. Such results will also provide a benchmark for performance of future vaccines and validate the epitope studied here as an important target.

## Methods

**Cloning of the 85RF45.1 antibody from hybridoma.** Sequencing and cloning of the rat hybridoma immunoglobulin genes were performed by LakePharma, Inc., Belmont, CA. Sequencing included cDNA synthesis, PCR with rat heavy and light chain primers, subcloning of correct-sized bands into TOPO vectors, and DNA sequencing of individual amplified clones. Native 85RF45.1 and chimeric 85RF45.1

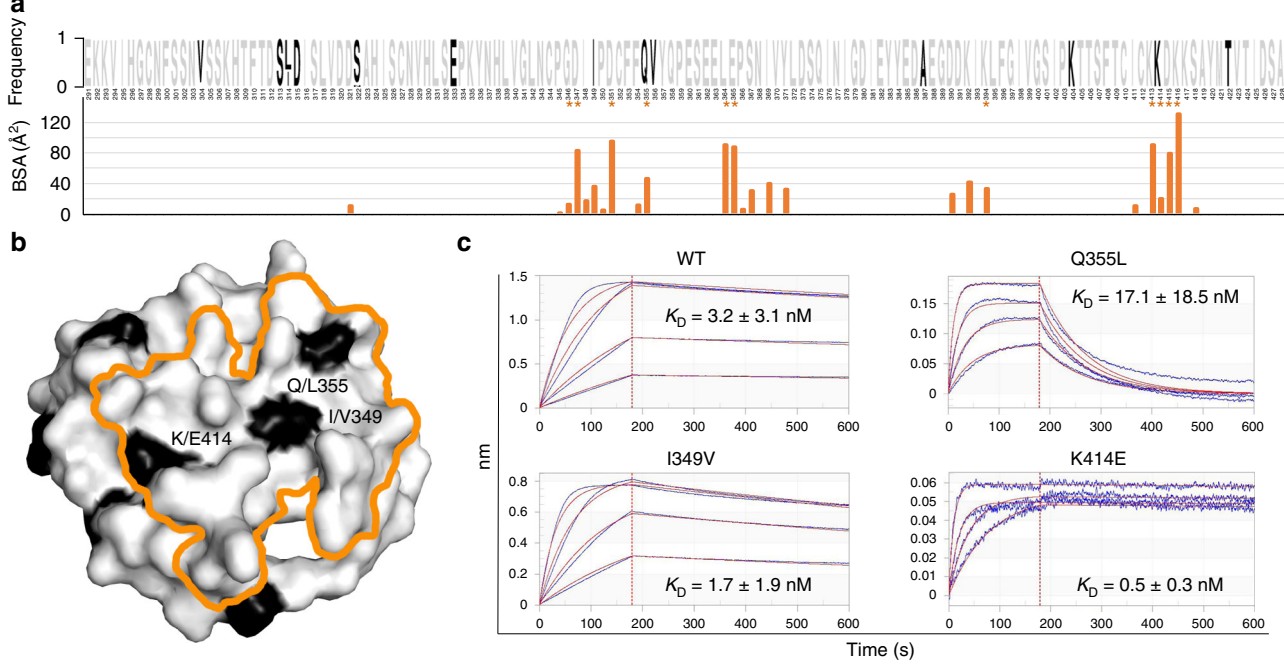

**Fig. 4** Pfs48/45 sequence polymorphisms in the 85RF45.1 epitope. **a** Weblogo[58] representation of sequence variability in Pfs48/45 6C. Sequences were compiled from the NCBI, PlasmoDB, and Uniprot databases. Invariant and polymorphic residues are colored gray and black, respectively. The buried surface area of each Pfs48/45 residue contacted by mAb 85RF45.1 as determined by PISA[59] is shown below. An asterisk denotes a Pfs48/45 residue that forms an H-bond or salt bridge with mAb 85RF45.1. **b** Polymorphism mapped onto the Pfs48/45 surface colored according to **a**. The 85RF45.1 epitope is outlined in orange. **c** Binding affinity of 85RF45.1 Fab to Pfs48/45 6C constructs with point mutations representative of sequence polymorphisms in the 85RF45.1 epitope. Blue lines are representative of raw data, whereas red curves represent global fitting. The different sensograms correspond to Fab concentrations of 250, 125, 62.5, and 31.3 nM. $K_D$'s are indicated with standard deviation and derive from at least two independent measurements

(fused to human IgG1 constant regions) antibodies were cloned into LakePharma's high expression mammalian vector system and transient production was performed in HEK293 cells (Thermo Fisher Scientific). Starting at 20 h, and throughout the transient transfection production, antibody titers were measured (Octet QKe, ForteBio). Cultures were harvested at day 5 and antibodies in the conditioned media were purified using MabSelect SuRe Protein A resin (GE Healthcare) for validation.

**Humanization of mAb 85RF45.1.** Humanized antibodies were designed by creating multiple hybrid sequences that fuse select parts of the parental antibody sequence with the human framework sequences. Humanness scores, representing how human-like the antibody variable region sequence is, were calculated according to Gao et al.[42]. Based on the method employed, heavy chain scores of 79 or above were considered human-like; for lambda light chains, a score of 84 or above was considered humanlike. Full-length antibody genes for all nine heavy/light chain pairs were constructed by cloning the synthesized variable region sequences into expression vectors that contained the human IgG1 or lambda constant region genes.

**Expression and purification of Fabs and antigens.** The VL and VH regions of mAb TB31F and mAb 85RF45.1 were cloned into the pHLsec expression vector[43] upstream of human Igλ and CH1 constant regions, respectively. Fabs were expressed in HEK293F cells (Thermo Fisher Scientific) and purified by affinity chromatography using a HiTrap LambdaSelect column (GE Healthcare), followed by ion exchange (MonoS; GE Healthcare) and gel filtration (Superdex 200 Increase 10/300 GL; GE Healthcare) chromatography. The NF54 [https://www.ncbi.nlm.nih.gov/protein/AHA91190.1] Pfs48/45 6C gene was codon-optimized for expression in mammalian cells, was cloned in the pHLsec expression vector[43], and was purified by affinity chromatography (HisTrapFF; GE Healthcare) followed by gel filtration chromatography (Superdex 200 Increase 10/300 GL; GE Healthcare). The R0.6C protein was produced as previously described[30].

**Complex formation of Pfs48/45 6C with 85RF45.1 and TB31F Fabs.** To obtain high yields of antigen–antibody complexes, plasmids for the TB31F and 85RF45.1 Fab heavy and light chains were mixed with the Pfs48/45 6C plasmid in a ratio of 60:30:40 and transfected in HEK293S cells (GnT I$^{-/-}$) (ATCC CRL-3022). 85RF45.1 Fab-Pfs48/45 6C and TB31F Fab-Pfs48/45 6C complexes were purified by affinity chromatography (HisTrapFF; GE Healthcare), treated with Endo H, and purified by cation exchange chromatography (MonoS; GE Healthcare).

**Crystallization and structure determination.** Hanging or sitting drop crystallization trials were set up using commercial sparse-matrix screens (Rigaku TOP96 and Anatrace MCSG 1) and hits were subsequently refined. 85RF45.1 and TB31F Fabs were set up at 8 mg/mL, while 85RF45.1 Fab-Pfs48/45 6C and TB31F Fab-Pfs48/45 6C complexes were set up at 5 and 6 mg/mL, respectively. Crystals of the Fabs were obtained in the following crystallization conditions. 85RF45.1 Fab: 0.2 M sodium thiocyanate, pH 6.9, 20% (w/v) PEG 3350; TB31F Fab: 0.1 M HEPES, pH 7.5, 10% (v/v) 2-propanol, 20% (w/v) PEG 4000. Diffraction-quality crystals of 85RF45.1 Fab-Pfs48/45 6C and TB31F Fab-Pfs48/45 6C complexes were obtained by microseeding in 0.1 M Bis-Tris, pH 5.5, 0.1 M ammonium acetate, 17% (w/v) PEG 10000 and 0.1 M Bis-Tris, pH 6.0, 22% (w/v) PEG 3350, 0.2 M NaCl, respectively. All crystals were cryo-protected by the addition of 20% (v/v) glycerol prior to flash-freezing in liquid nitrogen. Data were collected at the Canadian Light Source 08ID-1 beamline. Data processing and scaling was performed using XDS[44]. Phaser[45] was used for molecular replacement using Fabs from our internal database as starting models. AutoBuild[46] was used to build an initial model of Pfs48/45 6C, followed by manual building in Coot[47]. Structure refinement was carried out with phenix.refine[48]. Illustrations were generated using UCSF Chimera[49] and The PyMOL Molecular Graphics System, Version 2.0 Schrödinger, LLC. Software used in this project was curated by SBGrid[50]. Representative electron density for the structures is shown in Supplementary Figure 7.

**Sequence polymorphism analysis.** Deposited sequences for the Pfs48/45 6C fragment were obtained from the NCBI, PlasmoDB, and Uniprot databases. Sequence alignment was performed using Clustal Omega[51].

**BLI-binding studies.** Binding kinetics studies for the two Fabs (85RF45.1 and TB31F) interacting with Pfs48/45 6C were conducted using an Octet RED96 system (Pall ForteBio). For Fab-Pfs48/45 6C binding experiments, Pfs48/45 6C monomers were loaded onto Ni-NTA biosensors followed by transfer into Fab-containing wells. After the association steps were conducted, biosensors were then dipped into 1× kinetics buffer (1× PBS, pH 7.4, 0.002% (v/v) Tween-20, 0.01% (w/v) BSA) to measure dissociation. Kinetics analysis was performed using the ForteBio data analysis software. For experiments to screen humanized variants of mAb 85RF45.1, the binding study involved immobilization of the antibodies onto anti-human Fc (AHC) biosensors followed by dipping into the R0.6C antigen in solution.

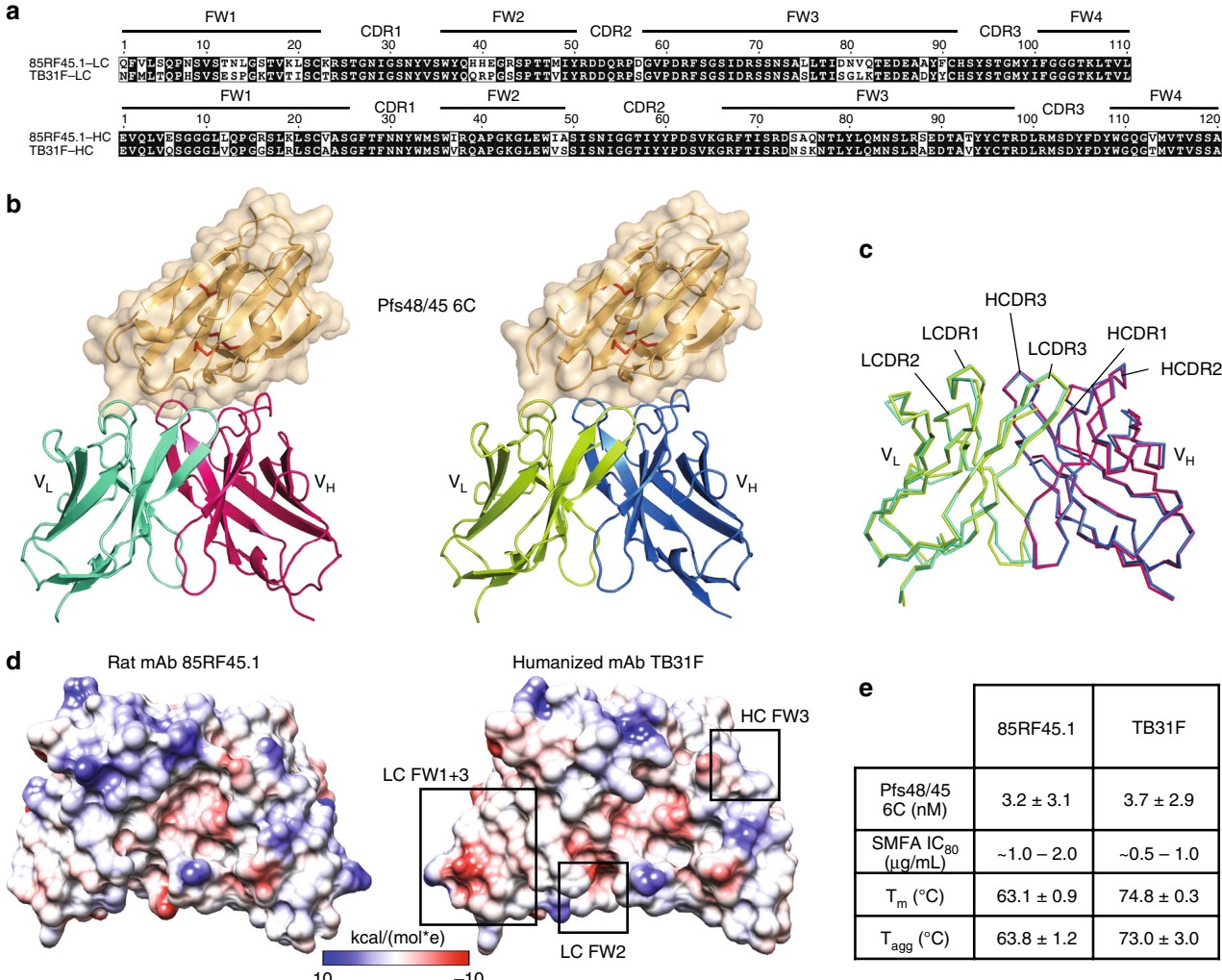

**Fig. 5** Humanization of transmission-blocking mAb 85RF45.1. **a** Sequence alignment between rat-derived mAb 85RF45.1 and its humanized counterpart mAb TB31F. CDRs and framework regions (FW) are defined according to the Kabat convention[60]. **b** Comparison of the 85RF45.1Fab-Pfs48/45 6C and the TB31F Fab-Pfs48/45 6C crystal structures. Pfs48/45 is shown as an orange surface and secondary structure cartoon, whereas Fabs are represented as secondary structure cartoons. Only the Fab variable region is shown for clarity. **c** Superposition of the variable regions of the unliganded 85RF45.1 Fab (teal and magenta) and the unliganded TB31F Fab (green and blue) crystal structures. **d** Coulombic electrostatic surface coloring of the variable regions of the unliganded 85RF45.1 Fab (left) and the unliganded TB31F Fab (right). Areas of major differences in surface electrostatics are boxed. **e** Biophysical and functional parameters between 85RF45.1 and TB31F. Errors are standard deviations derived from at least three independent experiments

**Melting and aggregation temperatures**. Melting and aggregation temperatures for 85RF45.1 and TB31F Fabs were measured on a UNit system (Unchained Labs). The $T_m$ was obtained by measuring the barycentric mean fluorescence, and the $T_{agg}$ was determined from static light scattering measurements at a 266 nm wavelength during a temperature ramp from 20 to 95 °C with 1 °C increments. The equilibration time was set to 60 s before each measurement. The concentration of both samples was 1.0 mg/mL. Measurements were made in 100 mM sodium acetate, pH 5.6; 100 mM HEPES, pH 7.2; and 100 mM Tris, pH 9.0; all containing 150 mM NaCl. Measurements were made in triplicates, averaged, and standard errors were calculated by the UNit analysis software.

**SMFA**. SMFA experiments were conducted as previously described[52] using *Anopheles stephensi* mosquitoes that were reared and maintained at Radboudumc, and transgenic *Pf* NF54 (NF54-HGL) gametocytes that express luciferase[53]. Briefly, 3–5-day-old mosquitoes were fed on a glass mini-feeder system containing 0.27 mL of cultured *Pf* gametocytes mixed with human serum and monoclonal antibody. For each feed, two mini-feeders containing gametocytes and human serum only were used as a negative control. On day 9 after infection, four pools of five mosquitoes were collected for each condition, stored overnight at −20 °C, homogenized by bead beating, and luciferase activity was quantified by measuring relative light units (RLU)[52]. The percentage transmission reduction activity (TRA) was calculated as $100 \times [1 − (RLU\ test\ sample)/(RLU\ control\ group)]$. The best estimate and 95% confidence intervals from two separate feeds were calculated using a negative binomial model as previously described[54] without zero inflation.

## Data availability

The crystal structures reported in this manuscript have been deposited in the Protein Data Bank: 6E62, 6E63, 6E64, 6E65. The authors declare that all other data supporting the findings of this study are available within the article and its Supplementary Information files are available from the authors upon request.

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

## Acknowledgements

We thank M. Theisen for providing us with the R0.6C reagent for screening humanized mAbs. The excellent work by Marga van de Vegte-Bolmer, Rianne Stoter, Wouter Graumans, and Geert-Jan van Gemert for conducting Standard Membrane Feeding Assays is highly appreciated. We thank Shwu-Maan Lee, Emily Locke, and Christian Ockenhouse for assistance in designing experiments and for the analysis of data relating to the selection of mAb TB31F. We thank Ashley Birkett and Jeffery Smith for helpful comments on the manuscript. This work was funded by PATH's Malaria Vaccine Initiative under grant OPP1108403 from the Bill & Melinda Gates Foundation. The production of mAb TB31F was supported in part by the Intramural Research Program of the NIAID, NIH. X-ray diffraction experiments were performed using beamline 08ID-1 at the Canadian Light Source, which is supported by the Canada Foundation for Innovation, Natural Sciences and Engineering Research Council of Canada, the University of Saskatchewan, the Government of Saskatchewan, Western Economic Diversification Canada, the National Research Council Canada, and the Canadian Institutes of Health Research. This research was undertaken, in part, thanks to funding from the Canada Research Chairs program (to J.-P.J.).

## Author Contributions

Experimental conception and design: C.R.K. and J.-P.J.; data acquisition: P.K., A.S., M.M.J., and K.M.; analysis of data: P.K., A.S., M.M.J., K.M., R.W.S., C.R.K., and J.-P.J.; design of humanized mAb TB31F: A.L. and J.L.; design of selection criteria and identification of TB31F as the lead: M.J.M., V.L.P., C.R.K., and R.W.S.; drafting the article or revising it critically for important intellectual content: P.K., A.S., M.M.J., R.W.S., C.R.K., and J.-P.J.

## Additional information

**Competing interests:** The authors declare no competing interests.

