## [Peer Review File · Nature Communications]

Reviewers' comments:

Reviewer #1 (Remarks to the Author):

Malaria transmission-blocking vaccines (TBVs) have been regarded as one of the essential strategies towards malaria elimination. However, there have been only a few TBV candidates under development (Pfs25, Pfs48/45, Pfs230, etc.). Of them, Pfs48/45 was considered to be a promising TBV candidate because anti-Pfs48/45 mAb 85RF45.1 potentially inhibits parasite transmission to mosquito vectors by binding to epitope I in Pfs48/45 6C domain. In this study, the Authors structurally delineate potent epitope I and show how mAb 85RF45.1 recognizes an electronegative surface with nanomolar affinity. Pfs48/45 polymorphisms are rare for residues involved at the binding interface. More importantly, humanization of rat-derived mAb 85RF45.1 conserved the mode of recognition and activity of the parental antibody, while also improving its thermostability. I think the results of this study will provide the TBV community very important information for developing effective TBVs based on the Pfs48/45. All the works were carefully designed, clearly presented, and the manuscript is well written. I have the following comments for the improvement of this manuscript.

Comments:

Major

They clearly demonstrated the importance of glycosylation at N303 for the correct folding of the recombinant Pfs48/45 6C expressed in HEK293. However, more important evidence that the TBV community requires is whether the recombinant glycosylated Pfs48/45 6C induce potent transmission-blocking antibodies or not. If yes, they will be able to bring their recombinant Pfs48/45 6C to further development pipeline. If they already have the data, please add them to this manuscript.

Minor

1) Lines 280-285

Why co-transfection of both mAb and Pfs48/45 6C plasmids required? Is this method essential to obtain mAb-Pfs48/45 & C complex? After the independent transfection, can the mAb (H+L) and Pfs48/45 6C form complex? Please add reason why the Authors used co-transfection method.

2) The Pfs48/45 6C expression in HEK293 cells required "codon optimized" gene or not?

3) Figs 4B, S1

"The 85RF45.1 epitope is outlined in orange."

The orange color is unclear. Please highlight this line for better visibility.

4) Line 231

"its high-affinity binding" to "its high-affinity binding"

Reviewer #2 (Remarks to the Author):

This paper describes the crystal structure of a potent transmission blocking antibody in complex with its epitope, the C-terminal domain of Pfs48/45, a target for transmission blocking vaccine.

The paper is well written and should be of interest to the malaria vaccine field.

There is not much background on the in vivo protective function of the antibody, which may be useful (although may not exist). Since the authors describe a humanized version of the antibody as a potential tool for prophylaxis in human, it will be nice to have more details on the basis if possible to make this more relevant.

The polymorphism analysis should be done on a larger number of sequences.

Some minor comments:

Describe what is the Z score for the structural similarity (where does it come from?)

Remove the electron density mesh in Fig 1B, does not really add (hard to see and make it more confusing). Move to Supplemental.

Reviewer #3 (Remarks to the Author):

This is the first report of the crystal structure of a Pfs48/45-specific malaria transmission blocking monoclonal antibody and a recombinant Pfs48/45 protein. It extends our understanding of the structure of the 6-cys motif domain in this important malaria vaccine candidate and provides insights into the design of future recombinant vaccine candidates that are directed against a specific transmission blocking epitope. Based on this work they develop a humanized mAb and demonstrate similar binding characteristics.

I would appreciate it if the authors would address the following questions/comments in the results/discussion.

Does the HEK295 expressed Pfs48/45 6C induce transmission-blocking antibodies?

Does the humanized mAb also bind to the different polymorphic Pfs48/45 sequences?

They demonstrate rat mAb binding to Pfs48/45 with single aa substitutions in the epitope, but have not formally tested for the ability of either mAb to block field isolates containing these polymorphisms in a SMFA. This caveat should be included in the discussion.

The interactions of TB31F-Pfs48/45 and 85RF45.1-Pfs48/45 should be compared directly in one table.

The luciferase data for the controls and four pools of mosquitoes from both feeds used to calculate the transmission-reducing activity in figure S4 should be included in a supplemental table.

Response to Reviewers' comments:

Reviewer #1 (Remarks to the Author):

Malaria transmission-blocking vaccines (TBVs) have been regarded as one of the essential strategies towards malaria elimination. However, there have been only a few TBV candidates under development (Pfs25, Pfs48/45, Pfs230, etc.). Of them, Pfs48/45 was considered to be a promising TBV candidate because anti-Pfs48/45 mAb 85RF45.1 potentially inhibits parasite transmission to mosquito vectors by binding to epitope I in Pfs48/45 6C domain. In this study, the Authors structurally delineate potent epitope I and show how mAb 85RF45.1 recognizes an electronegative surface with nanomolar affinity. Pfs48/45 polymorphisms are rare for residues involved at the binding interface. More importantly, humanization of rat-derived mAb 85RF45.1 conserved the mode of recognition and activity of the parental antibody, while also improving its thermostability. I think the results of this study will provide the TBV community very important information for developing effective TBVs based on the Pfs48/45. All the works were carefully designed, clearly presented, and the manuscript is well written.

We thank the reviewer for these encouraging comments.

I have the following comments for the improvement of this manuscript.

Comments:

Major

They clearly demonstrated the importance of glycosylation at N303 for the correct folding of the recombinant Pfs48/45 6C expressed in HEK293. However, more important evidence that the TBV community requires is whether the recombinant glycosylated Pfs48/45 6C induce potent transmission-blocking antibodies or not. If yes, they will be able to bring their recombinant Pfs48/45 6C to further development pipeline. If they already have the data, please add them to this manuscript.

We agree with the reviewer that a direct implication of our work will be in the design and development of an antigen capable of inducing potent transmission-blocking antibodies. The importance of the glycosylation of N303 for recombinant expression of Pfs48/45 6C in the HEK 293 recombinant system was clearly demonstrated in our work. Efforts are ongoing to further stabilize the antigen and develop immuno-focusing strategies based on the structures we describe. These new Pfs48/45 6C reagents will enable testing in immunization studies whether potent transmission-blocking antibodies can be preferentially elicited. These experiments have not yet been conducted, represent a considerable amount of work over several months and are therefore beyond the scope of the current work.

Minor

1) Lines 280-285

Why co-transfection of both mAb and Pfs48/45 6C plasmids required? Is this method essential to obtain mAb-Pfs48/45 &C complex? After the independent transfection, can the mAb (H+L) and Pfs48/45 6C form complex? Please add reason why the Authors used co-transfection method.

The reviewer correctly notes that co-transfection of the Fab and antigen was performed to obtain material for co-crystallization. This strategy was simply to increase yields, as is now noted in the Materials and Methods section: “To obtain high yields of antigen-antibody complexes, plasmids for the TB31F and 85RF45.1 Fab heavy and light chains were mixed with the Pfs48/45 6C plasmid in a ratio of 60:30:40 and transfected in HEK293S cells (GnT I^{-/-}).” As shown in BLI experiments (Figures 4 and 5, and Supplementary Figure 3), individually-expressed antigen and antibody molecules are fully capable of interacting with nanomolar affinity.

2) The Pfs48/45 6C expression in HEK293 cells required “codon optimized” gene or not?

The Pfs48/45 6C gene was indeed codon-optimized for expression in mammalian cells, as is now noted in the Materials and Methods: “The Pfs48/45 6C gene was codon-optimized for expression in mammalian cells, was cloned in the pHlsec expression vector⁴², and was purified by affinity chromatography (HisTrapFF column, GE Healthcare) followed by gel filtration chromatography (Superdex 200 Increase 10/300 GL column, GE Healthcare).”

3) Figs 4B, S1

“The 85RF45.1 epitope is outlined in orange.”

The orange color is unclear. Please highlight this line for better visibility.

The highlight of the 85RF45.1 epitope is now of slightly different colors and has been made with a thicker outline to improve clarity.

4) Line 231

“its high-affinity affinity binding” to “its high-affinity binding”

We thank the reviewer for catching this typo, which has now been corrected.

Reviewer #2 (Remarks to the Author):

This paper describes the crystal structure of a potent transmission blocking antibody in complex with its epitope, the C-terminal domain of Pfs48/45, a target for transmission blocking vaccine.

The paper is well written and should be of interest to the malaria vaccine field.

We thank the reviewer for these encouraging comments.

There is not much background on the in vivo protective function of the antibody, which may be useful (although may not exist). Since the authors describe a humanized version of the antibody as a potential tool for prophylaxis in human, it will be nice to have more details on the basis if possible to make this more relevant.

The reviewer correctly points out that there is little information available regarding the in vivo protective efficacy of 85RF45.1 or TB31F. In fact, this is a major reason of the current planning to advance TB31F to clinical testing where the most relevant in vivo data will be obtained. As pointed out in the Discussion, such an experimental clinical trial will establish a concentration of antibody concentration capable of blocking transmission from human to mosquito and this will be a key guide to both vaccine and possible mAb development as interventions. These plans are outlined at the end of the Discussion section: “mAb TB31F now offers a unique opportunity to test the efficacy of a Pfs48/45-based transmission-blocking intervention in humans. The mAb has entered good manufacturing production, and preclinical safety testing. Phase I trials are planned to assess safety and tolerability. Pending a positive outcome, this will be followed by efficacy testing with the goal of establishing the level of mAb required to block parasite transmission from human to mosquito, as determined by direct skin feeding (infection prevalence endpoint). Comparison of these data to those from the laboratory-based SMFA, which employs an infection intensity (as opposed to infection prevalence) endpoint and is routinely used to inform go/no-go criteria in pre- and early-clinical testing, will enable ‘back-validation’ of the SMFA to better inform its use as an early development stage-gate for measuring the efficacy of candidate vaccines and antibodies. Such results will also provide a benchmark for performance of future vaccines and validate the epitope studied here as an important target.”

The polymorphism analysis should be done on a larger number of sequences.

The polymorphism analysis was performed on all available Pfs48/45 6C sequences deposited in public databases (Materials and Methods: “Deposited sequences for the Pfs48/45 6C fragment were obtained from the NCBI, PlasmoDB and Uniprot databases.”) Additional data mining of Pfs48/45 sequences from the field will be critical to better understand past, present and future polymorphisms in potent transmission-blocking epitopes. These future studies will now be anchored in a structural understanding of Pfs48/45 and the potent 85RF45.1 epitope from our studies.

Some minor comments:

Describe what is the Z score for the structural similarity (where does it come from?)

The Z-score definition used by the Dali server is described in Holm et al., 2016. We have now added this reference at the end of the sentence in the Results that mentions Z-scores: “The three proteins with highest structural similarity with Pfs48/45 6C are schizont surface protein Pf12 (Z-score 12.9), merozoite surface protein Pf41 (Z-score 12.0), and Toxoplasma gondii sporozoite-specific SAG protein (Z-score 10.5) (Figure 2C)³¹.”

Remove the electron density mesh in Fig 1B, does not really add (hard to see and make it more confusing). Move to Supplemental.

We agree with the reviewer that the layered electron density meshes of different colors used to highlight the Pfs48/45 6C cysteines was hard to see. The depiction of Fig. 1B has therefore been updated to remove the electron density mesh around the cysteines. We maintained the overall electron density mesh around Pfs48/45 6C, which is a way to show its surface attributes using primary data, but reduced the thickness of its mesh to improve clarity.

Reviewer #3 (Remarks to the Author):

This is the first report of the crystal structure of a Pfs48/45-specific malaria transmission blocking monoclonal antibody and a recombinant Pfs48/45 protein. It extends our understanding of the structure of the 6-cys motif domain in this important malaria vaccine candidate and provides insights into the design of future recombinant vaccine candidates that are directed against a specific transmission blocking epitope. Based on this work they develop a humanized mAb and demonstrate similar binding characteristics.

I would appreciate it if the authors would address the following questions/comments in the results/discussion.

Does the HEK295 expressed Pfs48/45 6C induce transmission-blocking antibodies?

As mentioned above in response to Reviewer 1, a direct implication of our work will be in the design and development of an antigen capable of inducing potent transmission-blocking antibodies. The mammalian-cell expressed Pfs48/45 6C we report here will enable testing in immunization studies whether potent transmission-blocking antibodies can be preferentially elicited. These experiments are planned but have not yet been conducted, represent a considerable amount of work over several months and are therefore beyond the scope of the current work.

Does the humanized mAb also bind to the different polymorphic Pfs48/45 sequences?

As described in Figure 5, the paratope of humanized TB31F is nearly identical to the precursor antibody 85RF45.1, and as such would be expected to accommodate Pfs48/45 sequence polymorphisms in the same way. To confirm this point, we performed additional BLI experiments now reported in Supplementary Figure S3B. This data shows that TB31F interacts with Pfs48/45 of different sequence polymorphisms (I/V349, Q/L355 and K/E414) with similar binding kinetics as 85RF45.1. We have now added a sentence in the Results to describe these additional experiments: “TB31F also retained the ability of 85RF45.1 to recognize Pfs48/45 with sequence polymorphisms in its epitope (Supplementary Figure 3B).”

They demonstrate rat mAb binding to Pfs48/45 with single aa substitutions in the epitope, but have not formally tested for the ability of either mAb to block field isolates containing these polymorphisms in a SMFA. This caveat should be included in the discussion.

The reviewer correctly points out that it would be desirable to associate strain sequence variation and SMFA activity in order to better understand the breadth of activity of the 85RF45.1 antibody. We do not have, or have access to, clones carrying the documented sequence polymorphisms. The isolation and preparation of strains for directly testing SMFA activity using strains with the specific sequence variation is a major endeavor and outside the scope of the current manuscript. What is planned and underway is to explore whether Direct Membrane Feeding Assay (DMFA) testing can identify some evidence of variation in activity when blood is used from infected donor with divergent field strains. Again, this is a major endeavor outside the scope of the manuscript, but will be the matter of future reports. We now include a mention of this caveat in the Discussion: “We also now reveal that its epitope is largely conserved: mAb 85RF45.1 can accommodate all sequence polymorphisms previously reported and still bind with nanomolar affinity. However, whether this antibody can block field isolates containing these polymorphisms remains to be determined. mAb 85RF45.1 represents a unique tool to assess the efficacy of antibodies to reduce Pf transmission and malaria incidence in affected communities.”

The interactions of TB31F-Pfs48/45 and 85RF45.1-Pfs48/45 should be compared directly in one table.

Supplementary Tables 2 and 3 have now been merged to allow direct comparison of their interactions.

The luciferase data for the controls and four pools of mosquitoes from both feeds used to calculate the transmission-reducing activity in figure S4 should be included in a supplemental table.

This data has been added and is now reported in Supplementary Table 3.

REVIEWERS' COMMENTS:

Reviewer #1 (Remarks to the Author):

The authors have responded to all of the comments appropriately.

Reviewer #3 (Remarks to the Author):

The authors have responded to revisions requested. However, one additional change is needed that I must have missed before.

On Pg 10 line 207. A reference is needed that indicates the Pfs48/45 plays a role in Pf gamete fusion or other function. The reference given is just for adhesion. Thanks and sorry for the oversight.

It will be very interesting to see how the clinical work progresses with the humanized mAb.

Response to Reviewers' comments:

Reviewer #1 (Remarks to the Author):

The authors have responded to all of the comments appropriately.

We thank the Reviewer for the positive assessment of our revisions.

Reviewer #3 (Remarks to the Author):

The authors have responded to revisions requested.

We are glad that the Reviewer found our revisions adequate.

However, one additional change is needed that I must have missed before.

On Pg 10 line 207. A reference is needed that indicates the Pfs48/45 plays a role in Pf gamete fusion or other function. The reference given is just for adhesion. Thanks and sorry for the oversight.

We thank the Reviewer for accurately noting the divergence between the claim and its reference. Consequently, we have adjusted the sentence describing Pfs48/45 function to more accurately reflect the work referenced: “[...] associated with its function in male gamete fertility¹³ [...]”

13. van Dijk MR, *et al.* A central role for P48/45 in malaria parasite male gamete fertility. *Cell* 104, 153-164 (2001).

It will be very interesting to see how the clinical work progresses with the humanized mAb.

We also look forward to these results.